# NASPY: Automated Extraction of Automated Machine Learning Models

**Xiaoxuan Lou[1], Shangwei Guo[2], Jiwei Li[3,4], Yaoxin Wu[1] and Tianwei Zhang[1]**
[1]Nanyang Technological University, [2]Chongqing University, [3]Zhejiang University, [4]Shannon.AI
{xiaoxuan001, wuyaoxin, tianwei.zhang}@e.ntu.edu.sg, swguo@cqu.edu.cn, jiwei_li@shannonai.com

## Abstract

We present NASPY, an end-to-end adversarial framework to extract the network architecture of deep learning models generated by Neural Architecture Search (NAS). Existing model extraction attacks mainly focus on conventional DNN models with very simple operations, or require heavy manual analysis with lots of prior knowledge. In contrast, NASPY introduces seq2seq models to automatically identify novel and complicated operations (e.g., separable convolution, dilated convolution) from hardware side-channel sequences of model inference. We design two models (RNN-CTC and transformer), which can achieve only 3.2% and 11.3% error rates for operation prediction. We further present methods to recover the model hyper-parameters and topology from the operation sequence. With these techniques, NASPY is able to extract the complete NAS model architecture with high fidelity and automation, which are rarely analyzed before.

## 1 Introduction

Recently Automated Machine Learning (AutoML) has attracted lots of attention from the machine learning community, as it can significantly simplify the development of machine learning pipelines with high efficiency and automation. One of the most popular AutoML techniques is Neural Architecture Search (NAS) (Zoph & Le, 2016; Elsken et al., 2019), which can automatically generate high-quality Deep Neural Networks (DNNs) for a specified task. It enables non-experts to produce machine learning architectures and models which can outperform hand-designed solutions.

From the adversarial perspective, this paper aims to design new attacks to *steal* the architectures of black-box NAS models. This is known as *model extraction attacks*, which could cause severe consequences: (1) searching a good architecture with NAS is an energy- and time-consuming process. Hence the produced architecture is naturally considered as an important intellectual property, and stealing it can lead to copyright violation and financial loss (Hong et al., 2020). (2) Extracting the model architecture can facilitate other black-box attacks, e.g., data poisoning (Demontis et al., 2019), adversarial examples (Ilyas et al., 2018), membership inference (Shokri et al., 2017).

One solution of model extraction is to remotely query the target model and recover the architecture based on the responses (Oh et al., 2019). However, such attack requires large computation cost and can only be applied to simple neural networks[1]. Then we turn to a more promising solution: hardware attacks (e.g., cache side-channel, bus snooping). Essentially, when a DNN model executes the inference task on a computer, it leaves architecture-dependent footprints on the low-level hardware components, which could be captured by an adversary to analyze and recover the high-level architecture details. These techniques can give very fine-grained information, and have been utilized by prior works (Yan et al., 2020; Hua et al., 2018; Naghibijouybari et al., 2018; Hu et al., 2020) to extract the architectures of conventional DNN models. However, there are several challenges when we apply such attack techniques to extract NAS architectures. (1) These works can only handle simple operations in conventional models, while failing to analyze new operations introduced by NAS (e.g., separable convolution, dilated convolution). (2) Some works need complicated manual analysis with prior knowledge of the victim model. For instance, (Yan et al., 2020) requires the information of the victim model family, and can only extract variants of generic architectures. (Naghibijouybari et al., 2018) needs to know the layer type ahead.

---

[1]It takes 40 GPU-days to recover a 7-layer architecture with a simple chained topology (Oh et al., 2019).

To the best of our knowledge, there is only one work (Hong et al., 2020) focusing on the extraction of NAS models, which is not very practical or general. (1) This work mainly monitors the API traces in the high-level deep learning library, which requires the attacker and victim to share the same library with exactly the same version. This is not practical since users may use different libraries, especially their customized ones. (2) It needs the accurate dimension estimation to predict the layer type and model topology, which is hard to obtain in the real world. (3) The NAS model considered in this work is too small and simple, which cannot well represent state-of-the-art NAS techniques.

We propose `NASPY`, a learning-based framework for automated extraction of NAS architectures with high efficiency and fidelity. We make several contributions to overcome the above limitations. First, we exploit cache side-channel techniques to monitor the low-level BLAS library. Hence, the framework can be applied to different platforms regardless of the high-level deep learning libraries (Tensorflow, Pytorch, or other customized libraries). This cannot be achieved in (Hong et al., 2020). Second, we model the extraction attack as a sequence-to-sequence problem, and design new deep learning models to predict the model operation sequence automatically. This does not require the tedious manual analysis, as conducted in (Yan et al., 2020). Meanwhile, the models are able to predict new sophisticated operations in NAS, which are missing in (Yan et al., 2020; Hu et al., 2020). Third, we propose a new analysis method to precisely recover the exact hyper-parameters without any prior knowledge. In contrast, previous works can only estimate a possible range of hyper-parameter values (Yan et al., 2020; Hu et al., 2020). Finally, we design strategies to reconstruct the model topology and extract the complete architecture for different scenarios and adversarial goals.

We perform extensive experiments to demonstrate the effectiveness of `NASPY`. Our identification model can predict the operation sequences of different NAS methods (DARTS (Liu et al., 2018), GDAS (Dong & Yang, 2019) and TE-NAS (Chen et al., 2021)) with an error rate of 3.2%. Our hyper-parameter prediction can achieve more than 98% accuracy. The framework also demonstrates high robustness against random noise introduced by the complex and dynamic hardware systems. The source code of `NASPY` is available at `https://github.com/LouXiaoxuan/NASPY`.

## 2 Background

**Neural Architecture Search (NAS).** This promising autoML technology can systematically generate a good network architecture for a given task and dataset (Zoph & Le, 2016; Elsken et al., 2019). It defines a *search space* as the scope of neural networks in consideration, from which it finds the best architecture with different types of *search strategies* (Zoph & Le, 2016; Real et al., 2019; Liu et al., 2018). To reduce the search complexity and cost, a neural network is decomposed into multiple *cells* (Chen et al., 2021). A cell can be represented as a Directed Acyclic Graph, where each edge represents a connection between two neural nodes that is associated with an operation selected from a predefined operation set (Dong & Yang, 2020). A NAS architecture normally has two types of cells: a *normal cell* computes the feature maps and a *reduction cell* reduces the spatial size. Multiple normal cells construct a block, and multiple blocks are interconnected by reduction cells to form the final model. Compared to traditional neural networks, NAS adopts some sophisticated operations among the neural nodes, like dilated convolutions and separable convolutions.

**Hardware attacks.** Following the previous works (Yan et al., 2020; Hong et al., 2020; Hu et al., 2020), we aim to exploit some hardware attacks to perform the extraction of NAS models. Specifically, we consider the following two attacks.

*Cache side-channel attacks:* CPU caches are introduced between the CPU cores and main memory to accelerate the memory access. Since the attacker can share the same cache with the victim, he can reveal the behavior pattern of the victim from the contention on the usage of cache lines. In this paper, we adopt FLUSH-RELOAD, which is also used in (Yan et al., 2020; Hong et al., 2020) for model extraction. The adversary leverages the shared low-level BLAS library to infer the victim model, and is able to obtain the sequence of its critical operations, e.g., the matrix multiplication.

*Bus snooping attack:* data traffic between the processor and memory system is achieved through a *bus*, which sends data to or loads data from specific addresses. Hence, by observing the memory traffic through the bus, the attacker can obtain the memory address traces of the victim model, which further reveals the connections between model layers. In this paper, we use the bus snooping technique (Huang et al., 2014) to monitor the read/write addresses of each model layer, which is also

adopted in (Hu et al., 2020). The adversary can only observe the data addresses and cannot access the data passing through buses, which allows `NASPY` to work even when the model is encrypted.

**Sequence-to-sequence learning.** Seq2seq learning is raising increased attention in the machine learning community, and becomes quite popular for different tasks like speech recognition (Amodei et al., 2016), machine translation (Neubig, 2017), image captioning (Islam et al., 2019), question answering (Palasundram et al., 2020), etc. Three models are mostly used for seq2seq learning: Recurrent Neural Network (RNN), Connectionist Temporal Classification (CTC), and attention models (Transformer). In this paper, we aim to use seq2seq learning for automated model extraction from the monitored memory activities. We design an RNN-CTC model and a Transformer model to recover the structure operations of NAS models.

## 3 FRAMEWORK OVERVIEW

### 3.1 THREAT MODEL

**Adversary's goal.** Given a victim model $M$ constructed from NAS, the adversary aims to recover a similar network architecture as $M$, without searching it with large cost and the original dataset. We consider two types of goals following (Jagielski et al., 2020): (1) *accuracy extraction*: the adversary aims to reproduce a network architecture, which can give similar model accuracy as the victim model; (2) *fidelity extraction*: the adversary wishes to recover the same architecture and hyper-parameters as the victim one.

**Adversary's capability.** We consider two practical scenarios for extracting model architectures. For each scenario, we assume the attacker only knows the target model is from NAS, without any other prior knowledge, e.g., the model family, layer type, NAS method, high-level deep learning library.

• *Remote attack*: model extraction in this scenario is adopted in (Yan et al., 2020; Hong et al., 2020). The attacker can launch his malicious program on the same machine with the victim model. Although these programs are isolated by the OS or hypervisor, the attacker can still exploit the cache side-channel technique to monitor the victim's low-level executions, e.g., the sequence of matrix multiplication events. Based on the side-channel sequence, the attacker is able to perform accuracy extraction of the victim model.

• *Local attack*: this scenario is considered in (Hu et al., 2020), where the attacker can physically access the machine running the victim model. In addition to launching the cache side-channel attack to retrieve the operation sequence, the attacker can also launch the bus snooping attack to monitor the memory bus and PCIe events. With such memory address traces, the attacker can achieve fidelity extraction of the victim model and recover the exact model topology.

### 3.2 ATTACK OVERVIEW

Before describing our `NASPY` framework, we need to understand the workflow of the model inference process. As shown in the left diagram of Figure 1, given a DNN model, a deep learning library (e.g., Tensorflow, Pytorch) is used to process the computational graph of the model, and convert the model architecture into sequences of connected layer operations. Then these operation sequences are sent to low-level computation libraries for acceleration, such as the BLAS library (e.g., *OpenBLAS*) for GEneral Matrix Multiplication (GEMM), and the mathematical library (e.g., *libm*) for activation functions. Those computations will be executed on the hardware platform. By monitoring the hardware activities using cache side-channel and bus snooping techniques, the attacker can observe the event sequences and memory address traces for the model inference process.

**Overview of `NASPY`.** The adversary's task is to automatically and precisely recover the NAS model architecture from the captured sequences of hardware activities. Figure 1 shows the workflow of our `NASPY` framework. It consists of three steps, as described below.

First, it translates the event sequences from cache side-channel attacks to the operation sequences (i.e., input of the low-level computation library). We model this as a seq2seq problem, and design two deep learning models (RNN-CTC and Transformer) to achieve this prediction. Second, it identifies the values of hyper-parameters in the layer operations recovered from the first step. Previous works (Hu et al., 2020) can only estimate a range of these values based on the dimension size of layer

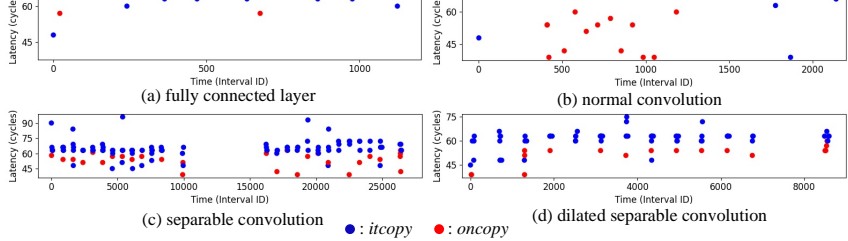

Figure 1: Workflow of our model extraction framework.

input and output. In contrast, NASPY can precisely reveal the exact values of the hyper-parameters from the translated operation sequence. Third, NASPY reconstructs the model topology and obtains the complete architecture. For the *accuracy extraction* attack, the attacker can randomly select a model topology, and assign the recovered operations with the hyper-parameters to it. Our experiment results show that the corresponding model can give similar accuracy as the victim one. For the *fidelity extraction* attack, the attacker needs to recover the exact model topology. He can leverage the information in the memory address trace from the bus snooping attack to construct the architecture.

## 4 DETAILED DESIGN

### 4.1 OPERATION SEQUENCE IDENTIFICATION

The first step is to predict the operation sequence from the hardware event sequence. Past works (Yan et al., 2020; Hong et al., 2020) require manual analysis for such translation. In contrast, we propose to leverage a seq2seq deep learning model to achieve this task automatically.

**Dataset formulation and preprocessing.** Specifically, the event sequence $x$ is a time-series of length $T$, where each frame is a vector of event features. In this paper, we capture the occurrence of *itcopy* and *oncopy* APIs in OpenBLAS, which are used to load matrix data for GEMM. Note that our framework is also generalized to other BLAS libraries, such as Intel MKL. Hence, a frame is denoted as $x_i = (I_i, O_i, T_i)$, where $T_i$ is the time interval from the last monitoring moment, and $I_i$ and $O_i$ are binary values to denote whether *itcopy* and *oncopy* are called during this interval. $T_i$ is determined by the monitoring granularity when collecting side-channel information.

The operation sequence $y$ contains $N$ operations performed by the victim model. To comprehensively cover novel neural architectures, we consider all the common operations used in state-of-the-art NAS methodologies: *fully connected layer (FC)*, *normal convolution (Conv)*, *dilated convolution (DConv)*, *separable convolution (SConv)*, *dilated-separable convolution (DSConv)*, *pooling (Pool)*, *identity (Skip)*, *zeroize*. It is easy to integrate other operations into NASPY if necessary.

Figure 2 shows the event sequences of four representative operations, where the blue and red nodes denote the occurrences of *itcopy* and *oncopy*. We observe that different operations have distinct event patterns, giving us opportunities to predict the model operations from the side-channel leakage. However, there are a couple of challenges to design seq2seq models for this task. We perform the following data preprocessing methods to overcome these challenges.

Figure 2: Event sequences of four representative operations in NAS models.

1. *Input downsampling and label extending.* An event sequence $x$ of a NAS model is extremely long ($> 5$ million frames) because of the high monitoring frequency. In contrast, an operation sequence $y$ usually has only hundreds of operations. It is infeasible to directly use existing seq2seq learning models to handle such length gap. To resolve this challenge, we perform downsampling for $x$ and extend labels in $y$. Specifically, for the event sequence $x$, we only keep the frames whose $I_i$ or $O_i$ is 1. We also update $T_i$ in these active frames as the time interval from the last function access. This can significantly reduce the sequence length, while preserving the critical information. For

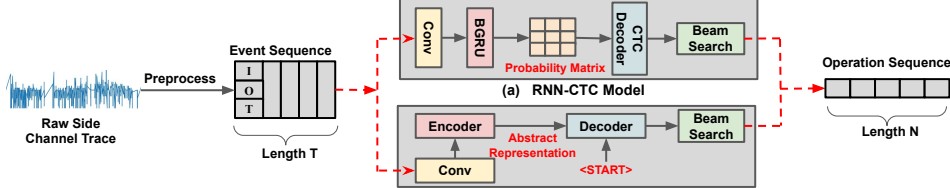

Figure 3: Procedure of operation sequence identification.

the operation sequence $y$, we decompose the single label of a complex operation into a series of sub-operation labels. For instance, a *SConv* operation is composed of several *Conv* sub-operations (Figure 2(c)). So instead of directly labeling this operation as *SConv*, we extend it as multiple *Conv* labels, the number of which is determined by the channel size. Such preprocessing can remarkably decrease the difficulty of training the seq2seq model.

2. *Inter-operation context.* Not every operation in $y$ has the corresponding event in $x$. For instance, *Pool* and *Skip* never invoke GEMM computations. This makes it difficult to predict such operations. We propose to leverage the execution latency and inter-operation context for identification, since their values are different in these operations. Specifically, we introduce two interval labels into the operation sequence: $I_\Gamma$ denotes the interval between two operations while $I_\gamma$ denotes the interval between the decomposed sub-operations within an operation. Such two labels reflect the context switch for inter- and inner-operations, which hence helps the seq2seq model distinguish adjacent operations in the sequence with high accuracy. Experiment results in Section 5.1 show that the consideration of inter-operation context brings large performance improvement.

3. *Time normalization.* When the victim model runs on different platforms, although the occurred events (i.e., $I_i$ and $O_i$) keep constant, the time interval $T_i$ between the frames will be different, due to the varied execution latency in hardware. This can restrict the generalization of our seq2seq model. To overcome this challenge, we perform normalization over $T_i$ in the captured side-channel trace, and use the relative intervals to train the model, which are similar across various platforms as they are determined by the algorithm behaviors.

4. *Data augmentation.* Different from NLP and CV tasks, the raw side-channel data can contain lots of random noise from the hardware activities. This is due to the complex system optimization and run-time dynamics (Hu et al., 2020). Such noise can decrease the identification accuracy. To improve the robustness of the seq2seq model against noise, we further perform data augmentation following Park et al. (2019), which simply cuts out random blocks of consecutive time and feature dimensions. In this paper, we just mask these blocks with the fixed value 0.

**Seq2seq model designs.** We propose two kinds of models for predicting the operations from the side-channel trace. The first one is an **RNN-CTC model**. Recently the combination of RNN and CTC decoders is commonly used in sequence modeling problems. Figure 3(a) shows the architecture of this model for identifying the operation sequence $y$ from the event sequence $x$. Given that $x$ only contains three features in each frame, we first introduce a convolution layer to learn more features. Then an RNN layer is used to propagate information through this sequence. We adopt the Bidirectional Gated Recurrent Unit (BGRU) as the RNN module, which can enable better long-term memory and fully leverage the past and future contexts. The output of the RNN layer is the probability distribution of all the operation labels for each frame, which is further fed into the CTC decoder. While it is difficult to align the operation sequence with the event sequence that has a various length, the CTC decoder can skip the alignment by introducing a "blank" label. Finally, the operation sequence $y$ with the largest prediction probability is identified with the beam search.

The second one is a **transformer model.** Proposed by Vaswani et al. (2017), transformers have broken multiple AI task records and pushed the state of the art. Hence, we also show the possibility of operation identification with a transformer model, as shown in Figure 3(b). Transformer is an attention-based neural network with a typical encoder-decoder architecture. The encoder maps the input sequence into an abstract representation that contains all the learned features of that input. Then the decoder takes this abstract representation and predicts the next output step-by-step based on the previous output. The introduction of the attention mechanism enables the transformers to have extremely long term memory, which can focus on all the tokens generated previously. Similarly, we add a convolution layer before the encoder to learn more features from the event sequence.

## 4.2 HYPER-PARAMETER RECOVERY

Our second step is to extract the architectural hyper-parameter values of each operation. NASPY can precisely recover the exact hyper-parameters, rather than an estimated range in previous works (Hu et al., 2020; Yan et al., 2020).

**Convolutions.** Given that convolutions take a majority in a NAS model, we first discuss how to reveal hyper-parameters of different convolution operations, e.g., kernel size $R$, dilation $d$, channel size $C$, padding size $P$ and stride. Note that we assume the input size (i.e., width and height) has been identified from the analysis of the previous layer, while the initial input size of the model is publicly available. (1) The extended event labels in $y$ is integrated to compose the single label of the complex operation (e.g., *SConv* and *DSConv*). During this process, NASPY discovers the channel size $C$ by counting the number of event labels. (2) Both the kernel size $R$ and dilation $d$ can be directly recovered from the predicted operation labels (e.g., $3 \times 3$ *SConv* or $5 \times 5$ *DSConv*). NASPY can distinguish various convolutions with different kernel sizes, as they have different side-channel patterns and execution time. (3) Empirically, the padding size $P$ is normally set as $R/2$ to keep the input size and output size constant for possible residual connections. (4) The stride can be deduced from the cell type. Normal cells keep the spatial size unchanged while reduction cells would half the spatial size but double the channel size to maintain the dimension information. Hence, the stride is 1 for normal cells and 2 for reduction cells. The cell type is identified through the channel size $C$.

**Other operations.** We can also infer the hyper-parameters for other operations like the *FC* and *Pool* layers. The number of neurons in a *FC* layer can be reflected by the length of the event sequence, where a larger number of neurons leads to a longer sequence. Such length variance can be well learned by the seq2seq model, which discloses the number of neurons in the predicted labels. For the *Pool* layer, the relationship between the pooling size $R$ and padding size $P$ is $P = R/2$ in order to keep the spatial size among layers in the same cell.

## 4.3 MODEL TOPOLOGY RECONSTRUCTION

The final task is to extract the topology to obtain the model architecture. As a NAS model is built with cells, we also reconstruct the model topology in term of cells. First, the model macro skeleton is determined by analyzing the number and types of cells from the event sequence. It is intuitive to locate each cell, since they are separated by much larger time intervals due to some extra computations like concatenating and preprocessing. Then, we focus on the topology of each cell. Unlike conventional DNN models, the topology in NAS cells is not chained and sometimes even not regular. Hence, we cannot simply connect each extracted operation in the sequence to form the topology. Based on the attack scenarios and goals, the cell reconstruction can be done with different methods.

**Accuracy extraction.** To extract a model architecture with similar accuracy, the attacker can just use the remote side-channel attack to recover the operations and hyper-parameters, and then randomly choose a topology following the basic rule of NAS, i.e., each neuron node has two inputs. The experiment results in Section 5.3 verify the effectiveness of this strategy.

**Fidelity extraction.** As only the side-channel event sequence is not enough to achieve fidelity extraction, the attacker can adopt bus snooping to get the memory address trace to reveal the exact interconnections between layers. With the revealed model topology, the attacker can finally extract the complete model architecture.

## 5 EVALUATION

**Testbed.** NASPY is general for different deep learning and computation libraries. Without loss of generality, we adopt Pytorch (1.8.0) and OpenBLAS (0.3.13).

**Dataset construction.** We search model architectures with CIFAR10, and train model parameters over CIFAR10 and CIFAR100. Our method can be applied to tasks for other datasets as well. We first generate 10,000 random computational graphs of NAS models following the macro skeleton proposed in the popular benchmark NAS-Bench-201 (Dong & Yang, 2020), where each cell contains 4 nodes associated with 8 operations (each node has two inputs). The operation set follows the classical set in (Liu et al., 2018), which contains: *identity*, *zeroize*, *3×3* and *5×5 SConv*, *3×3* and

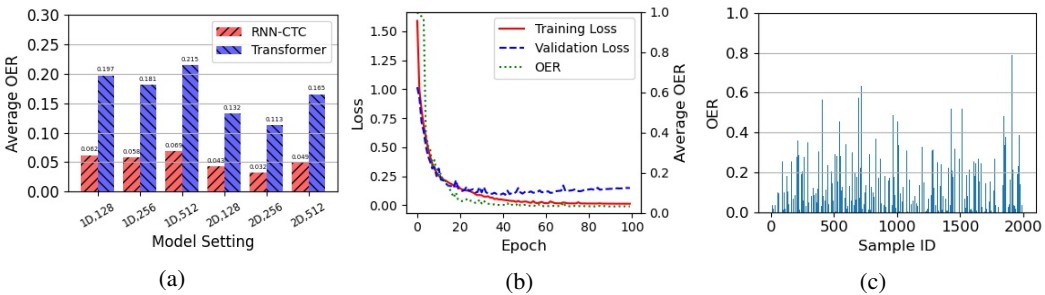

Figure 4: (a) Average OER of the two identification models with different configurations. (b) Loss and OER trend of the RNN-CTC model. (c) OER of the transformer model on validation samples

*5×5 DSConv*, *3×3 average pooling*, *3×3 max pooling*. Conventional operations (i.e., *Conv* and *FC*) are also included in our model.

Then we collect the corresponding side channel sequences using the FLUSH-RELOAD technique, which inspects the cache lines storing OpenBLAS functions (*itcopy* and *oncopy*) at a granularity of 2000 CPU cycles. We randomly select 80% of the sequences as the training set, and the rest as the validation set. The seq2seq models will be tested on novel NAS models generated by three state-of-the-art NAS methods: DARTS (Liu et al., 2018), GDAS (Dong & Yang, 2019) and TE-NAS (Chen et al., 2021). For each method, we search 10 NAS models with various initial seeds for testing.

### 5.1 Operation Sequence Identification

**Metrics.** Inspired by the evaluation metric WER (Word Error Rate) in NLP tasks, we propose Operation Error Rate to quantify the prediction accuracy. It is calculated as: $OER = D(y', y)/|y|$, where $D(y', y)$ is the edit distance between the predicted operation sequence $y'$ and ground-truth sequence $y$, and $|y|$ is the sequence length of $y$. A smaller OER implies higher identification accuracy.

**Prediction accuracy.** Given that our dataset is relatively small, we only use one layer of BGRU in the RNN-CTC model, and one layer of encoder and decoder in the transformer model. Figure 4(a) shows the average validation OER of two models with different structure settings. Both *1D Conv* and *2D Conv* are considered to extract more features from the input, and the model dimension size (i.e., $d_{rnn}$ of the BGRU layer and $d_{model}$ of the transformer) is chosen among [128, 256, 512]. We see that while all the configurations can achieve very high accuracy, the best one is: *2D Conv* and $d_{model} = 256$, where both models give the lowest OER: 3.2% for RNN-CTC and 11.3% for the transformer. The trend of the loss and average OER during the training of RNN-CTC are depicted in Figure 4(b). We observe that the loss and OER decrease dramatically within the first 20 epochs and reach convergence at epoch 40. In contrast, the transformer model performs relatively worse. Figure 4(c) shows the OER of each validation sample predicted by the transformer, which is higher and more unstable. It is because the length of the preprocessed side-channel event sequence (normally 5000 frames) is still too long for the transformer, which requires tremendous GPU and memory resources, making the training more difficult.

Finally, we test the above two models on the NAS models generated by DARTS, GDAS and TE-NAS[2], and the results are shown in Table 1. From the table, both of the RNN-CTC and transformer models can well predict the opera-

| Model | Average | DARTS | GDAS | TE-NAS |
|---|---|---|---|---|
| RNN-CTC | 0.032 | 0.035 | 0.021 | 0.038 |
| Transformer | 0.113 | 0.151 | 0.115 | 0.094 |

Table 1: Testing OER for three NAS methods.

tion sequence from the side-channel leakage of three types of NAS models.

**Effectiveness of inter-operation context.** We adopt the *inter-operation context* technique to handle the missing events of some operations (Section 4.1). To evaluate its effectiveness, Figure 5 compares the prediction error rates of the two models with and without considering the context. Figure 5(a) shows the OER trend of the RNN-CTC model during training. We observe that without the inter-operation context, the OER decreases more slowly and converges at a higher value. Figure 5(b) depicts the fitting curves on the OER of validation samples for the transformer. Ignoring the context will lead to a huge performance penalty, which gives higher and less stable prediction error rates.

---

[2]We train these models using the open-sourced code from the authors.

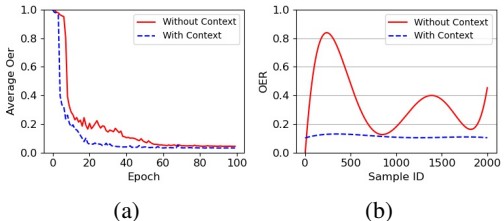
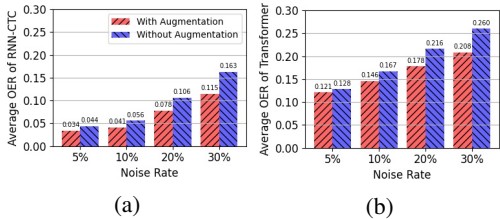

Figure 5: Inter-operation context testing. (a) OER trend of RNN-CTC. (2) OER of the transformer on validation samples.

Figure 6: Robustness versus different scales of noise. (a) OER of RNN-CTC. (b) OER of the transformer.

**Robustness against noise.** We further evaluate the robustness of our models against the noise. Figure 6 shows the OER of two models without and with the data augmentation technique, when the side-channel event trace contains different scales of random noise. First, we observe that our two models have strong robustness and give acceptable error rates under large amounts of noise. Second, data augmentation (masking rate = 0.1) can further improve the model robustness. With 30% random noise, the OER of RNN-CTC is 0.115 and the transformer is 0.208. Besides, data augmentation has better improvement for RNN-CTC than the transformer, as the transformer always considers the entire input, which weakens the effect of this technique.

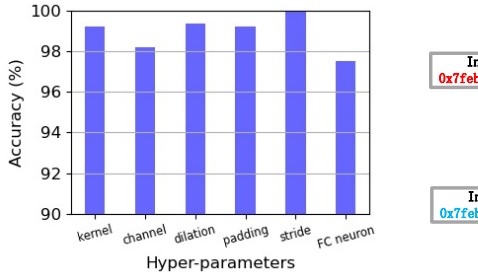
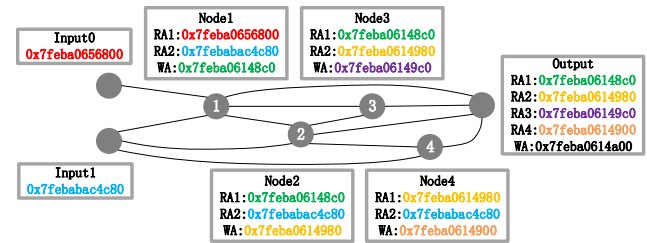

Figure 7: Recovery accuracy.

Figure 8: Memory address trace of a NAS cell.

## 5.2 HYPER-PARAMETER RECOVERY

NASPY can further extract the hyper-parameter values from the predicted operation sequence. As RNN-CTC performs better, we conduct hyper-parameter recovery based on the operation sequence from this model. The hyper-parameter prediction error is identical to the operation identification error (3.2%). Figure 7 shows the recovery accuracy of various hyper-parameters. For convolutions, the accuracy of each hyper-parameter can reach up to 98%. The kernel size and dilation are recovered directly from the predicted labels and have the highest accuracy. The padding size is computed from the kernel size with the same accuracy. The stride is inferred from the cell type, which can reach 100% accuracy, as normal and reduction cells are very distinguishable in terms of the channel size. The recovery accuracy of the channel size is relatively lower, because most errors in the predicted operation sequence are from the adding or deletion of events, while the channel size is revealed by counting the repetitions of events in a convolution. However, such slight accuracy drop can be easily compensated with post-analysis. For the number of neurons in *FC* layers, the recovery accuracy is 97.54%, as the side channel pattern of *FC* is shorter and simpler, making it hard to be distinguished sometimes. The pooling size cannot be directly reveal from the operation sequence, as the changing of pooling size does not lead to significant changes of the side-channel leakage pattern. So for this specific operation, we can determine its value empirically from the common values (i.e., 3 or 5).

## 5.3 MODEL TOPOLOGY RECONSTRUCTION

First, we consider the accuracy extraction attack. After the attacker identifies the operations and hyper-parameters, he can randomly choose a model topology to obtain the architecture, which can give close model

| Dataset | Original Model | Random Model with Same Operations | | | | |
|---|---|---|---|---|---|---|
| | | #1 | #2 | #3 | #4 | #5 |
| CIFAR 10 | 96.82 | 96.53 | 96.44 | 96.60 | 96.57 | 96.77 |
| CIFAR 100 | 81.07 | 80.95 | 80.16 | 80.33 | 80.90 | 80.56 |

Table 2: Accuracy (%) of random models on two datasets.

performance. To validate this, given a victim NAS model, we first extract its operation sequence and hyper-parameters. Then we randomly generate a computation graph that connects nodes in the cell, and associate the revealed operations to the graph sequentially. We train the model from this graph and test its prediction accuracy. Table 2 shows the results on CIFAR10 and CIFAR100 datasets with 5 different graphs. We can see that the randomly chosen topology can give very similar accuracy as the original model, where the accuracy drop is less than 1% for both datasets. We also check the performance of totally randomly constructed models (operations and topology): the average accuracy of 5 such models is 91.85% (CIFAR10) and 72.49% (CIFAR100), which is much lower than our extracted models. This shows the importance of operation and hyper-parameter recovery for accuracy extraction.

Second, we consider fidelity extraction. The memory address trace of the victim model can be obtained with bus snooping tools, e.g., HMTT-v4 (Huang et al., 2014). Figure 8 shows an example of the monitored address trace of a cell in the victim model, where `RA` is the *read* address and `WA` is the *write* address. If `RA` of one layer $a$ is the same as the `WA` of another layer $b$, we confirm that there is a connection from $b$ to $a$. By analyzing the consistency between `RA` and `WA` in the address traces, the connections between layers can be recovered precisely. We perform experiments on 6 NAS models in Table 2 and the results confirm that the model topology can be revealed with 100% precision. Combining the recovered operations and hyper-parameters, which may contain some small errors, we can finally reconstruct the complete architecture that is almost the same as the original one.

## 6 DISCUSSIONS

**Extracting other NAS models.** Our framework can be extended to attack other types of NAS models as well. For instance, some latest NAS techniques (e.g., Lin et al. (2021)) do not adopt the cell-based structure for searching. Since `NASPY` focuses on the recovery of operation sequences rather than cells, it is still effective to extract models from those NAS solutions. Another example is NAS-based RNN models. A RNN model generated by a NAS method from the search space only contains activation functions (Liu et al., 2018). It is noted that these functions cannot be observed from the GEMM trace. Instead, we can monitor other libraries (e.g., libm) to identify the functions. In the future, we will design seq2seq models and methods for extracting these models.

**Extracting standard DNN models.** Our framework can also be generalized to conventional standard DNN models. For models with chained topology (e.g., VGG), it is easy to extract their model architectures by first predicting the operation sequence and then just connecting them in sequence. However, for models with more complex topology (e.g., ResNet), after revealing the operation sequence, we need to perform fidelity extraction to reveal the exact model topology. If we know the family of the target model (assumed in Yan et al. (2020)), we can also reconstruct a similar model architecture by connecting the predicted operations as the template structure of the model family.

**Defense strategies.** There are several solutions that can possibly mitigate our extraction attacks. From the hardware perspective, oblivious RAM (Stefanov et al., 2013) was designed to hide the memory access pattern and thwart bus snooping attacks. Security-aware cache architectures (Werner et al., 2019; Qinhan et al., 2020) were proposed to reduce side-channel leakage. From the application perspective, we can obfuscate the model's execution behaviors by adding dummy operations or shuffling the operation orders (Hong et al., 2020). However, those solutions either require significant changes to the hardware, or add large computation overhead to the model inference. We will systematically evaluate those defenses, and design more efficient approaches as future work.

## 7 CONCLUSION

In this paper, we design `NASPY`, an end-to-end framework for automated extraction of DNN architectures from the NAS technology. `NASPY` adopts new deep learning models to identify model operation sequences from the side-channel event trace. With the identified operations, `NASPY` can further precisely recover the operation hyper-parameters and model topology for various scenarios. Compared to past works, `NASPY` can extract novel operations and architectures with higher automation and accuracy, bringing more severe threats to the protection of AI systems. We expect this study can raise the awareness of the community about the severity of model extraction attacks, and inspire researchers to come up with more secure NAS solutions to protect the architecture privacy.

ACKNOWLEDGMENTS

We thank the anonymous reviewers for their valuable comments. This project is in part supported by Singapore National Research Foundation under its National Cybersecurity R&D Programme (NCR Award NRF2018NCR-NCR009-0001), Singapore Ministry of Education (MOE) AcRF Tier 1 RS02/19, and NTU Start-up grant. Any opinions, findings and conclusions or recommendations expressed in this paper are those of the authors and do not reflect the views of National Research Foundation, Singapore

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

## A  DETAILS ABOUT GEMM IN OPENBLAS

BLAS realizes the matrix multiplication with the function *gemm*. This function computes $C = \alpha A \times B + \beta C$, where A is an $m \times k$ matrix, B is a $k \times n$ matrix, C is an $m \times n$ matrix, and both $\alpha$ and $\beta$ are scalars. OpenBLAS adopts Goto's algorithm (Goto & Geijn, 2008) to accelerate the multiplication using modern cache hierarchies. This algorithm divides a matrix into small blocks (with constant parameters P, Q, R), as shown in Figure 9. The matrix A is partitioned into $P \times Q$ blocks and B is partitioned into $Q \times R$ blocks, which can be fit into the L2 and L3 caches, respectively. The multiplication of such two blocks generates a $P \times R$ block in the matrix C.

Algorithm 1 shows the process of *gemm* that contains 4 loops controlled by the matrix size $(m, n, k)$. Functions *itcopy* and *oncopy* are used to allocate data and functions. *kernel* runs the actual computation. Note

| Library | Functions | Code Line |
|---------|-----------|-----------|
| OpenBLAS | Itcopy | kernel/generic/gemm_tcopy_8.c:78 |
|  | Oncopy | kernel/x86_64/sgemm_ncopy_4_skylakex.c:57 |

Table 3: Monitored code lines in OpenBLAS and Pytorch.

that the partition of $m$ contains two loops, $loop_3$ and $loop_4$, where $loop_4$ is used to process the multiplication of the first $P \times Q$ block and the chosen $Q \times R$ block. For different cache sizes, OpenBLAS selects different values of P, Q and R to achieve the optimal performance. Table 3 gives the monitored lines of the code in OpenBLAS 0.3.13.

---

**Algorithm 1:** *GEMM* in OpenBLAS

**Input:** matrice A, B, C; scalars $\alpha$, $\beta$
**Output:** $C = \alpha A \times B + \beta C$

1 **for** *j in (0:R:n)* **do** // Loop 1
2    **for** *l in (0:Q:k)* **do** // Loop 2
3       call *itcopy*
4       **for** *jj in (j:3UNROLL:j+R)* **do**
         // Loop 4
5         call *oncopy*
6         call *kernel*
7       **for** *i in (P:P:m)* **do** // Loop 3
8         call *itcopy*
9         call *kernel*

---

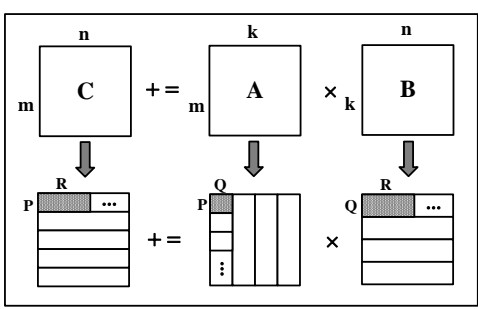

Figure 9: The procedure of GEMM.

## B  TRAINING DETAILS ABOUT IDENTIFIER MODELS

**CRNN+CTC model.** This model is comprised sequentially with one convolution layer $l_C$, one bi-directional GRU layer $l_R$ and one classifier $F$ with two FC layers. To evaluate the capability of $l_C$ on the feature learning, both 1d and 2d convolutions are adopt in experiments for comparison. Besides, to evaluate the performance of identifiers with different model sizes, three candidate dimensions of $l_R$ (i.e., 128, 256, 512) are considered. To train the model, we use CTC loss as the criterion to bypass the sequence alignment, and we use Adam optimization. The learning rate starts from $5\mathrm{e}{-4}$ and is scheduled following the OneCycleLR policy (Smith & Topin, 2019). The model is trained for 100 epochs, which takes 6.25 hours on one V100 GPU. The number of model parameters is 0.957 M.

**Transformer model.** The number of encoder layers and decoder layers in this model are both set as 1, and the dimension of model input and output $d_{model}$ is also chosen from the tuple (128, 256, 512). We employ $h = 8$ parallel attention layers, and the dimensions of the projected queries, keys and values are set as $d_k = d_v = d_{model}/h$. The inner-layer of the feed-forward layer is set with dimension $d_{ff} = 1024$. For the positional encoding and optimizer setting, we follow the implementation in (Vaswani et al., 2017). With above settings, we train the model with 100 epochs, which takes 20.41 hours on one V100 GPU. The number of model parameters is 1.872 M.

## C  DETAILS OF RANDOM GENERATED NAS MODEL GRAPH

Figure 10 shows an example of NAS cell structures with random chosen topology. In Section 5.3, we perform experiments on a original NAS model searched by GDAS and five models with the same operation but random topology. The details about these 6 models are shown in Table 4:

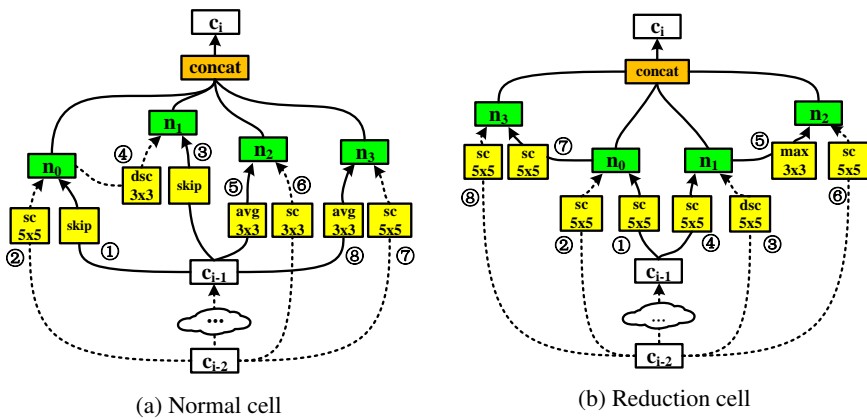

(a) Normal cell  (b) Reduction cell

Figure 10: An example of NAS cells with random topology.

| Model | Normal Cell | Reduction Cell |
|---|---|---|
| Original Model | $((skip, 1), (5 \times 5SConv, 0))$ | $((5 \times 5SConv, 1), (5 \times 5SConv, 0))$ |
| | $((skip, 0), (3 \times 3DSConv, 1))$ | $((5 \times 5DSConv, 0), (5 \times 5SConv, 2))$ |
| | $((3 \times 3avg\_pool, 0), (3 \times 3SConv, 3))$ | $((3 \times 3max\_pool, 1), (5 \times 5SConv, 2))$ |
| | $((5 \times 5SConv, 3), (3 \times 3avg\_pool, 2))$ | $((5 \times 5SConv, 3), (5 \times 5SConv, 1))$ |
| Random Model 1 | $((skip, 1), (5 \times 5SConv, 0))$ | $((5 \times 5SConv, 1), (5 \times 5SConv, 0))$ |
| | $((skip, 1), (3 \times 3DSConv, 2))$ | $((5 \times 5DSConv, 0), (5 \times 5SConv, 1))$ |
| | $((3 \times 3avg\_pool, 1), (3 \times 3SConv, 0))$ | $((3 \times 3max\_pool, 3), (5 \times 5SConv, 0))$ |
| | $((5 \times 5SConv, 0), (3 \times 3avg\_pool, 1))$ | $((5 \times 5SConv, 2), (5 \times 5SConv, 0))$ |
| Random Model 2 | $((skip, 0), (5 \times 5SConv, 1))$ | $((5 \times 5SConv, 0), (5 \times 5SConv, 1))$ |
| | $((skip, 0), (3 \times 3DSConv, 2))$ | $((5 \times 5DSConv, 0), (5 \times 5SConv, 2))$ |
| | $((3 \times 3avg\_pool, 2), (3 \times 3SConv, 3))$ | $((3 \times 3max\_pool, 3), (5 \times 5SConv, 0))$ |
| | $((5 \times 5SConv, 2), (3 \times 3avg\_pool, 3))$ | $((5 \times 5SConv, 4), (5 \times 5SConv, 0))$ |
| Random Model 3 | $((skip, 0), (5 \times 5SConv, 1))$ | $((5 \times 5SConv, 1), (5 \times 5SConv, 0))$ |
| | $((skip, 1), (3 \times 3DSConv, 2))$ | $((5 \times 5DSConv, 1), (5 \times 5SConv, 0))$ |
| | $((3 \times 3avg\_pool, 2), (3 \times 3SConv, 0))$ | $((3 \times 3max\_pool, 3), (5 \times 5SConv, 2))$ |
| | $((5 \times 5SConv, 4), (3 \times 3avg\_pool, 3))$ | $((5 \times 5SConv, 2), (5 \times 5SConv, 4))$ |
| Random Model 4 | $((skip, 1), (5 \times 5SConv, 0))$ | $((5 \times 5SConv, 1), (5 \times 5SConv, 0))$ |
| | $((skip, 0), (3 \times 3DSConv, 2))$ | $((5 \times 5DSConv, 0), (5 \times 5SConv, 1))$ |
| | $((3 \times 3avg\_pool, 1), (3 \times 3SConv, 2))$ | $((3 \times 3max\_pool, 1), (5 \times 5SConv, 0))$ |
| | $((5 \times 5SConv, 0), (3 \times 3avg\_pool, 2))$ | $((5 \times 5SConv, 4), (5 \times 5SConv, 3))$ |
| Random Model 5 | $((skip, 1), (5 \times 5SConv, 0))$ | $((5 \times 5SConv, 0), (5 \times 5SConv, 1))$ |
| | $((skip, 0), (3 \times 3DSConv, 2))$ | $((5 \times 5DSConv, 1), (5 \times 5SConv, 0))$ |
| | $((3 \times 3avg\_pool, 3), (3 \times 3SConv, 2))$ | $((3 \times 3max\_pool, 2), (5 \times 5SConv, 1))$ |
| | $((5 \times 5SConv, 2), (3 \times 3avg\_pool, 1))$ | $((5 \times 5SConv, 2), (5 \times 5SConv, 1))$ |

Table 4: Detailed cell structures of random NAS models. Each cell in a model contains 4 nodes (4 layers) and each node has two inputs, i.e., $(operations, node\ where\ input\ from)$

## D COMPARISON BETWEEN FIDELITY EXTRACTED NETWORK WITH THE ORIGINAL NETWORK

Figure 11 demonstrates an example of the raw memory address trace of a NAS cell. For the sake of simple understanding, we divide the whole trace into multiple blocks representing computational nodes in the cell. Each adjacent "read-write" pair in the trace means the data is first read from $addr_A$ and then is written back to $addr_B$, where $addr_A$ and $addr_B$ can be either the same or different according to the following usage of the data. By analyzing consistent between the read/write addresses, we can reveal the model topology with 100% precision. Figure 12 gives a comparison between the architecture obtained by the fidelity extraction and the original network. As we discussed in main text, while the monitored memory address traces allow the attacker to reveal the complete model topology precisely, the difference between the extracted model and the original one is only determined by the predicted operation sequence and hyper-parameters. In Figure 12 (a), two operations $3 \times 3DSC$ and $5 \times 5SC$ are marked as red, because the predicted operation sequences of them contain small errors, i.e., the repeated times of event labels (i.e., channel size) has slight deviations. But it is simple to fix them based on the model context. Hence, we can finally reveal the complete model architecture.

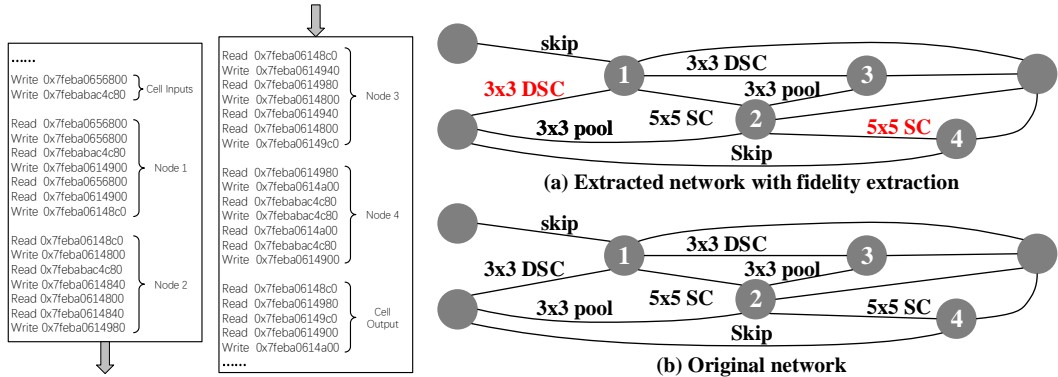

Figure 11: Raw memory address trace.  Figure 12: Comparison of fidelity extraction.

# E  EXAMPLE OF A RECURRENT CELL TRACE

Since machine learning privacy (Xu et al., 2020a;b) also involves Recurrent Neural Networks (RNN), we also try to exploit the availability of NASPY on RNN models. Figure 13a shows an example of a recurrent cell with DARTS. Figure 13b shows the corresponding side-channel trace for this cell. Each cluster is mapped to an activation function in the cell. We observe the side-channel trace has a much more observable pattern, where each cluster corresponds to a node in order. This makes it possible to predict the RNN operation sequence using our framework.

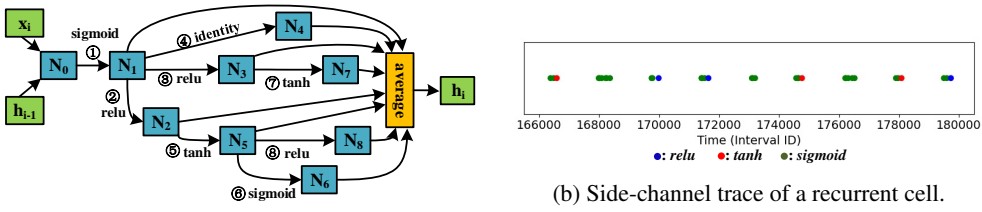

(a) Example of a recurrent cell.

(b) Side-channel trace of a recurrent cell.

Figure 13: An example of RNN NAS cells and trace.

