# OpenReview forum: "NASPY: Automated Extraction of Automated Machine Learning Models"
_ICLR.cc/2022/Conference — ICLR 2022 Spotlight_

### Official Review · Reviewer_L3iz · 2021-11-01

**Correctness:** 2
**Technical Novelty And Significance:** 3
**Empirical Novelty And Significance:** 3
**Recommendation:** 6
**Confidence:** 4

**Details Of Ethics Concerns:**

Not sure where to put this, so putting it here as it’s kind of bad academic practice. The authors exploit the supplementary material section by placing tables and figures in that section and then referring to them from the main body. This makes a mockery of having a page limit.

**Main Review:**

The area of NAS and attacks on NAS is an interesting area which is currently receiving quite a bit of attention. It is good to see work in this area.

Strengths:
- A more complete approach for attacking a NAS process.
- A well constructed and well-written piece of work.

Weaknesses:
- The paper makes some rather bold claims, which seem to stem from the fact that they assume all NAS approaches work in the same way and are based around the same building blocks. This assumption is not correct and would make this the approach far less effective.
- The predominant way to train Deep Learning and also for NAS is to use a GPU. However, the approach here seems to rely on the fact that all computations are performed on the CPU. This again would significantly reduce the effectiveness of the attack method.

Some comments on the text:
- The abstract and title are not good. Reading the abstract does not convey what this paper is about. Some text needs to be added to make it clear what domain this is in (i.e., trying to steal a network from a running NAS). Once you understand that the abstract starts to make sense.

- The paper is over the 9 page limit. You have cheated by placing figures and tables in the supplementary material and then referenced them from the main body. If this were to be allowed people would start submitting 9 pages of text and put all figures / tables / etc into the supplementary material. This makes a mockery of having a page limit.

- The footnote on page 1 needs either a citation or you need to provide your evidence to support this.

- Your idea of what NAS is and the underlying techniques used for it is rather restricted. You assume that NAS is a process of building up cells and that these cells are formed from standard units. However, there are many other NAS techniques out there which do not follow this model. For example differentiable NAS approaches, Zero-shot NAS (which doesn’t do any training) and NAS approaches which only partly train networks. These other approaches would not (I assume) be susceptible to your attack. You should clearly articulate what is required for your approach to work. This can be extracted by thinking through the whole process, but it should not be the readers role to do this.

- Likewise, the approach seems to assume that the whole process requires that you perform all of the NAS approach on the CPU. As the dominant approach for performing Deep Learning is to use a GPU there should be some discussion as to whether your approach would work in these cases. I’m guessing not.

- NAS rarely returns the ‘optimal network architecture’ - in most cases it returns a ‘good’ architecture, but to find the optimal you would need to perform all possible architectures - and even then you would only know it was ‘optimal’ within your search space.

- What do you mean by ‘a normal cell interprets the features’?

- There is an assumption that NAS introduces new more complex operations - this is not true. The NAS can only provide operations that it has been told to use.

- From the description of how you ‘grab’ the architecture from the NAS approach, I can’t see why this wouldn’t work if you were just training a network. Is this correct? If so say so, if not please explain why it wouldn’t work.

- What is a ‘proxy dataset’?

- A number of terms are used without being explained, including itchy, incopy, I_i, O_i

- Given that to ’steal’ a network you need to train your system and then run it against trace logs captured from the running NAS it would seem that this whole process is computationally expensive. Especially as you still need to train the stolen network. Given this it would seem essential that there is some comparison between the cost of training the NAS and training your approach and using it. It may be the case that it would be cheeper to just run the NAS yourself.

- Data augmentation is not clearly enough explained - can you add more details.

- What do you mean by “we first introduce a convolution layer to extract more features”. Convolution layers process the features they are given, but they do not create new features.

- There are some pretty big assumptions in this work. Such as P=R/2. This is not a requirement of performing NAS, so shouldn’t be a an assumption by you.

- The last two paragraphs in 4.3 seem to only contain information presented earlier. Remove and use for something better.

- You state that your approach can identify the hyper-parameters. However, it is only one type of hyper-parameters that you talk about. Those such as batch size, optimiser and learning rate don’t feature in this. I’m assuming that your approach does not cater for these - please make clear.

- In terms of batch size this seems to be an easy way to thwart your approach. There is no discussion of batch size in your work, so one would assume that the work does not consider it. Therefore changing the batch size would change the size of the operations and hence break your trained model.

- I’d want to see evidence to support the statement “Without loss of generality we adopt”. I can imagine cases where a different combination of libraries could throw your system off.

- More details on the results of the fidelity extraction attack should be presented. It’s very tiny covered at the moment.

- The conclusions are very short and don’t add a lot to the paper.


**Summary Of The Paper:**

The paper presents a technique for performing an attack on NAS approaches allowing the technique to identify the neural architecture which the NAS is honing in on.

**Summary Of The Review:**

The paper presents an approach to ’steal’ a DNN from a NAS training process. There are strong claims as to the applicability of the approach across all NAS approaches. However, the authors make an awful lot of assumptions as to how NAS works and the sorts of networks they can produce. If you factor in the NAS approaches they are not considering and the fact that the work seems to be focused on CPU training (which is not the norm) then the work has far more limited applicability.

Although the work may work for GPU, there is no discussion in the text of this therefore it is not possible to assess if it would work or not.

As someone who reviews for security venues too, I’d imagine this work would score higher at one of those venues.

---

> ### Author Response · Authors · 2021-11-15
> **Response to Reviewer L3iz**
>
> We thank the reviewer for your insightful review. We revise the paper according to your comments.
>
> **1. Reading the abstract does not convey what this paper is about**
>
> Thanks for pointing it out. In the revision, we have revised the abstract to make it more clear. The target of our work has been highlighted in the first sentence of the abstract.
>
> **2. Make a mockery of having a page limit by placing figures and tables in the supplementary material**
>
> We would like to clarify that our main paper is self-consistent and the figures and tables in the supplementary material are not indispensable for understanding our paper. We refer to these figures and tables because they can provide more details about our work for reviewers who have interest to explore further. We have removed these annotations in the revision.
> To better show our main paper is self-consistent, we would like to summarize the content of the four sections in supplementary material.
>
> 1. Section A gives more details about the GEMM implementation in the OpenBLAS library, while the background and functionality of our monitoring target, i.e., itcopy and oncopy functions, have been introduced in the main body.
>
> 2. Section B gives structure details and training details about two identifier models we used.
>
> 3. Section C gives the details about the random generated NAS models used for comparison in Section 5.3. We thought it was not necessary to put them in the main body.
>
> 4. We add a Section D to show more technical details about the fidelity extraction, like the monitored raw memory address trace and comparison between the extracted result and the original one.
>
> 5. Section E gives an example of NAS-generated RNN models, which provides a possible direction for exploiting our NASPY in the future.
>
> **3. Evidence needed to  support the assertion of the footnote on page 1**
>
> Thanks, we have added a citation [c], where the assertion comes from.
>
> **4. Articulate what is required for your approach to work, for example cell-based NASs. Can NASPY be used to extract Zero-shot NAS models?**
>
> Our method also works for the NAS methods that are not based on cells. This is because our pattern analysis performs at the level of operations, rather than cells. Our operation sequence identification can reveal each operation of victim models, regardless of whether the NAS models consist of cells. We focus on cell-based NASs because this approach is the mainstream and most NAS works follow the cell strategy, including the popular NAS benchmark, NASbench-201 . In terms of Zero-shot NAS (Zen-NAS), since it aims to improve the evaluation strategy while the used search space is similar with previous NAS works, we believe our NASPY can also extract Zen-NAS models. In the revision, we have added new discussions about the generalization of our NASPY in Section 6.
>
> **5. Why only extract CPU leakage for analysis. How about extracting models running on GPUs**
>
> Our method mainly focuses on the operation sequence analysis and architecture extraction. It is independent of the underlying platform and side-channel techniques. In GPUs, we can also use side-channel attacks to collect the runtime behaviors and then use NASPY to perform attacks. Collecting runtime behaviors on GPUs with side-channels have been demonstrated in [a][b]. We will consider applying our NASPY to GPU environments as an important direction for future work.
>
> **6. Hard to define and search the optimal network architecture**
>
> Yes, we totally agree. We have changed the description “optimal architecture” to “good architecture” in our revision.
>
> **7.What do you mean by ‘a normal cell interprets the features’**
>
> Thanks a lot for pointing this out. We intended to show that the normal cell is used to learn features from the last input. To avoid confusion, we have changed it to “a normal cell computes the feature maps” in our revision.
>
> **8. ‘new more complex operations’ is confusing**
>
> Thanks a lot for pointing this out. We just want to show that NAS models use some more complex operations (e.g., separate or dilated convs) than standard CNN models (e.g., VGG and ResNet). We have changed the description to “NAS adopts some sophisticated operations ....” in our revision.
>
> **9. From the description of how you ‘grab’ the architecture from the NAS approach, I can’t see why this wouldn’t work if you were just training a network. Is this correct? If so say so, if not please explain why it wouldn’t work.**
>
> Just training a network does not work. The goal of our attack is to steal a DNN model architecture from the victim, and cause some damages to him. For instance, it can lead to IP violation. It can also facilitate other black-box attacks, as introduced in the second paragraph of Section 1. Simply training a model has nothing to do with the victim, and will not bring any impact on the victim.

---

> ### Author Response · Authors · 2021-11-15
> **Response to Reviewer L3iz Part Two**
>
> **10. What is a ‘proxy dataset’?**
>
> A proxy dataset is a terminology in NAS. It refers to the dataset on which the NAS technique is performed to search for a good network architecture. The proxy dataset can be the same as the training dataset, or it can also be a smaller dataset for fast architecture searching. To avoid confusion, we have changed the description of “proxy dataset” to “dataset” in our revision.
>
> **11. A number of terms are used without being explained, including itchy, incopy, I_i, O_i**
>
> Thanks for pointing this out. In the revision, we give more explanations about these terms in Section 4.1 (the second paragraph).
>
> **12. It may be the case that it would be cheaper to just run the NAS yourself.**
>
> Please refer to #9. Our goal is to steal the architecture of the victim model, which could bring damages to the victim. Just running the NAS by ourselves cannot affect the victim.
>
> **13. More details about the data augmentation**
>
> Following your suggestion, we have added more details in the description of the data augmentation in the revision. Specifically, we follow the SpecAugment proposed in [d] and simply cut out random blocks of consecutive time and feature dimensions, i.e., masking these blocks with fixed value 0, to achieve the data augmentation.
>
> **14. Incorrect description: “we first introduce a convolution layer to extract more features”**
>
> Thanks for your correction, we have revised the description as : “we first introduce a convolution layer to learn more features”.
>
> **15. P=R/2 should not be an assumption by you**
>
> We agree. P=R/2 is a hidden feature in technical implementations, not an assumption in NAS. It is an empirical setting and commonly used in real implementations. To be more precise, we have modified our description as “Empirically, the padding size $P$ is normally set as $R/2$ to keep the input size and output size constant for possible residual connections ” .
>
> **16. The last two paragraphs in Section 4.3 can be removed.**
>
> Thanks for your advice. We rewrite this part to make it more concise.
>
> **17. Clarify what the hyper-parameters are.**
>
> The hyper-parameters refer to the properties of model architectures, e.g. kernel size and channel size. We do not consider the training details, like learning rate, batch size, etc. In the revision, we have clarified that the hyper-parameters are  “architectural hyper-parameters” in section 4.2 to avoid confusion.
>
> **18. Changing the batch size would change the size of the operations and hence break your trained model.**
>
> We would like to clarify that the batch size $N$ would only affect the computational complexity of our NASPY, as the side-channel leakage pattern of each operation would repeat for N times. Our attack still works because the analysis methodology is the same. When the batch size changes, we just need to merge these repeated operations and pay attention to the adjacent operations that are  the same in origin. Besides, for both the remote and local attack scenarios, we can both control the batch size of the query inputs to the victim model. Therefore, our trained model would not be broken.
>
> **19. Evidence to support the statement “Without loss of generality we adopt”**
>
> Our NASPY utilizes the cache leakage of  BLAS libraries that can be obtained using Cache Telepathy [e]. For each BLAS library,  the operation patterns would not change by varying the deep learning libraries. Therefore, our attack would have similar performance under different software environments.
>
> **20. More details on the results of the fidelity extraction attack should be presented.**
>
> Thanks for your advice.  We have added more details about fidelity extraction in Section 5.3.
>
> **21.The conclusions are very short and don’t add a lot to the paper.**
>
> We have polished the conclusion in the revision
>
> [a] Naghibijouybari, Hoda, et al. "Rendered insecure: GPU side channel attacks are practical." ACM SIGSAC Conference on Computer and Communications Security. 2018.
>
> [b] Naghibijouybari, Hoda, et al. "Side channel attacks on gpus." IEEE Transactions on Dependable and Secure Computing (2019).
>
> [c] Oh, Seong Joon, Bernt Schiele, and Mario Fritz. "Towards reverse-engineering black-box neural networks." Explainable AI: Interpreting, Explaining and Visualizing Deep Learning. 2019. 121-144.
>
> [d] Park, Daniel S., et al. "Specaugment: A simple data augmentation method for automatic speech recognition." arXiv preprint arXiv:1904.08779. 2019.
>
> [e] Yan, Mengjia, Christopher W. Fletcher, and Josep Torrellas. "Cache telepathy: Leveraging shared resource attacks to learn DNN architectures." USENIX Security Symposium. 2020.

---

> > ### Comment · Reviewer_L3iz · 2021-11-28
> > **Thanks for the feedback**
> >
> > This has answered a lot of my questions. However, there does still seem to be contradictions. You claim in 9 that your approach won't work if you're just training a network. However, you also claim in 4 that your approach will work for Zero-Shot NAS, which if you take the true definition of this is "identify a network without doing any training and then train it"...

---

> > > ### Author Response · Authors · 2021-11-29
> > > **Thanks for the comments**
> > >
> > > We would clarify that our claims in 4 and 9 are actually not contradicted. In response 4, “Just training a network does not work” means that training an independent model does not work under the threat model of DNN model extraction attacks, which aims to steal a victim’s proprietary model rather than getting a good model architecture. Our approach reveals the model operation sequence based on the side-channel leakage monitored during the **model inference**, which is totally independent from the model training. Hence, our approach is effective for DNN models searched and identified from Zero-Shot NAS. We will give more clarification about this point in the revision.

---

### Official Review · Reviewer_pLvS · 2021-11-03

**Correctness:** 4
**Technical Novelty And Significance:** 3
**Empirical Novelty And Significance:** 4
**Recommendation:** 8
**Confidence:** 3

**Main Review:**

# Significance
The topic in general and the contributions presented by the authors are highly relevant and of interest to a broad audience of ML researchers as well as practitioners.

# Novelty
To the best of my knowledge the main aspects proposed in the paper are novel contributions.

# Soundness
The paper seems to be sound. The claims are supported by a proper empirical study considering various scenarios.

# Writing
The paper is very well organized and the writing is very clear. Especially, explanations regarding rather technical or hardware-related aspects are very well done and understandable.

# Minor Comments
- Capitalization of headlines are inconsistent.
- The color schemes of figures should be adapted to be more inclusive (for red-green blindness). Especially, Figure 4 is very hard to read and distinguish between red and green for me.

# Questions
- What is the runtime and data complexity of the proposed approach?

**Summary Of The Paper:**

In the paper "NASPY: Automated Extraction of Automated Machine Learning Models" the authors propose an approach to extract the structure and parameters of a DNN that has been constructed via neural architecture search. In an empirical study the authors demonstrate the effectiveness of their approach and show that by making some assumptions they can indeed successfully extract the architecture of DNNs as well as their parameters.

**Summary Of The Review:**

All in all, the paper makes an important contribution to the area of security in machine learning. The proposed approach is able to successfully extract neural architectures and even their parameters, allowing for piracy of such models. Hopefully, this work will foster defense mechanisms in order to prevent such attacks. Therefore, I recommend to accept the paper.

---

> ### Author Response · Authors · 2021-11-15
> **Response to Reviewer pLvS**
>
> Thank you very much for your valuable comments. We revise the paper according to your comments. Please find our responses below:
>
> **1. Capitalization of headlines are inconsistent.**
>
> Thanks for pointing this out. We have fixed this error throughout the paper.
>
> **2.The color schemes of figures should be adapted to be more inclusive**
>
> This comment is much appreciated. We have redrawn the figures in our paper (Figure 4, 5, 6), where we use both different shapes and colors for different objects.
>
> **3.Runtime and data complexity of the proposed approach**
>
> Our approach is very efficient. We only need to perform model inference for one sample, and then predict the operation sequences with our pre-trained seq2seq model, and extract the architecture. The entire process is automated and we can extract a complex NAS model in the order of seconds.

---

### Official Review · Reviewer_haYd · 2021-11-03

**Correctness:** 4
**Technical Novelty And Significance:** 3
**Empirical Novelty And Significance:** 4
**Recommendation:** 8
**Confidence:** 3

**Main Review:**


I agree to accept this paper. The model is novel, the idea is well-motivated, and the experiments are compelling.

Pros:
1.	A novel, neat, and effective learning-based framework for automated extraction of NAS architectures with high efficiency and fidelity.
2.	Model the extraction attack as a sequence-to-sequence problem, and design new deep learning models to predict the model operation sequence automatically.
3.	A new analysis method to precisely recover the exact hyper-parameters without any prior knowledge.
4.	Design strategies to reconstruct the model topology and extract the complete architecture for different scenarios and adversarial goals.
5.	Comprehensive empirical results to show the effectiveness of the components.

Cons:
1.	The experiments mainly focus on extracting NAS models. However, in real-world scenarios, the variants of standard models (e.g., Resnet, Densenet) are widely used. Will NASPY still be effective on these models? Consider that, some hyper-parameters like the number of layers, channels, and stages may be varying.
2.	It would be helpful to briefly introduce the background hardware knowledge (e.g., the meaning of the itcopy, oncopy, and time interval). Many of the readers may be unfamiliar with these details.
3.	I think that the experiments may need to include a comparison between the architecture obtained by the fidelity extraction attack and the original network.

Discussions:
Would it be possible to extract the network architecture of detection and segmentation models? For example, the FPN or other detection/segmentation head.


**Summary Of The Paper:**

This paper presents an end-to-end adversarial framework to extract the network architecture obtained from Neural Architecture Search (NAS), named NASPY. Previous works focus on extracting conventional ConvNets with simple operations, while requiring heavy human analysis. In contrast, NASPY introduces a seq2seq network to automatically identify complicated operations from the hardware event sequence. In addition, both the value of hyper-parameters and the exact model topology can be obtained with NASPY. Extensive experiments based on DARTS, GDAS, and TE-NAS demonstrate the effectiveness of NASPY.

**Summary Of The Review:**

I agree to accept this paper. The model is novel, the idea is well motivated, and the experiments are compelling.

---

> ### Author Response · Authors · 2021-11-15
> **Response to Reviewer haYd**
>
> We thank the reviewer for your valuable review. We revise the paper according to your comments. Please find our responses below:
>
> **1. Will NASPY still be effective on the variants of standard models?**
>
> Yes, our NASPY can be generalized to the variants of DNN models (e.g., Resnet, ​​Densenet). The extraction of these models can also be modeled as a seq2seq problem, and the operations they adopt are already included in our sequential models. It is easy to extract these models using our NASPY. In the revision, we have added some discussions in Section 6 about the generalization of our approach to standard DNN models.
>
> **2. Introduce the background hardware knowledge**
>
> Thanks a lot for the good suggestion. In our revision, we have briefly introduced the explanation of itcopy/oncopy in Section 4.1 (the second paragraph). We have also added more background about these hardware-related terms in Appendix A for readers not familiar with this part.
>
> **3. Comparison between the architecture obtained by the fidelity extraction attack and the original network**
>
> Following your suggestion, in our revision, we have added more details about fidelity extraction in Section 5.3. An example of raw  memory address trace and a detailed comparison between the extracted network with the original one have also been added in Appendix D. With memory accessing addresses monitored by the bus snooping attacks, we can reveal the whole model topology with 100% precision. Hence, the difference between the extracted and original network architectures is only determined by the predicted operation sequence and hyper-parameters, which have been analyzed before in our paper.
>
> **4. The effectiveness of NASPY to extract the network architecture of detection and segmentation models**
>
> This is a very interesting idea. We think it is possible to extract the architecture of  the FNP model using NASPY. FPN utilizes a $1 \times 1$ convolution on each layer to reduce the channel dimensions and then a $3 \times 3$ convolution to generate the final feature map. Such fixed operation patterns can be easily recognized from the side-channel leakage sequence by our NASPY.  We will leave this as future work.

---

### Official Review · Reviewer_KWQS · 2021-11-03

**Correctness:** 3
**Technical Novelty And Significance:** 2
**Empirical Novelty And Significance:** 3
**Recommendation:** 6
**Confidence:** 2

**Main Review:**

1. I am not an expert of hardware-based attacks, but achieving 3.2% error rates for operation prediction seems quite impressive.

2. Due to the lack of knowledge on hardware attacks, I did not fully understand what assumptions are made to achieve this kind of low error rate. Could you clarify the assumptions of these attacks?

3. The authors claim that separable or dilated convs are complicated, but could you clarify what’s the key challenges to attach these operations?

4. The paper claims to be focused on NAS models, but it is unclear to me why the proposed approach is specific to NAS models? Since it is just based on model inference (without accessing the search process), there should be no difference whether the target model is a NAS model or not? Please clarify this.

**Summary Of The Paper:**

This paper proposes a hardware-based attack framework NASPY, which can extract NAS network architecture and hyper-parameters. The main idea is to use a seq2seq model to to identify operations from side-channel sequences.

**Summary Of The Review:**

1. Interesting approach and impressive results.
2. Need more clarifications for a few points.

---

> ### Author Response · Authors · 2021-11-15
> **Response to Reviewer KWQS**
>
> We thank the reviewer for your valuable review. We revise the paper according to your comments. Please find our responses below:
>
> **1. Clarify the assumptions of hardware-based attacks**
>
> Similar to other hardware-based attacks [a,b], we assume that the attacker can launch a malicious program on the target machines to passively monitor the cache access leakage of victim models.  This is a very standard and practical threat model. The low error rate is achieved by accurately analyzing the cache leakage pattern of each NAS operation. With the assumption and our pattern analysis, the attacker can construct a training dataset, train our well-designed sequence analysis models offline, and then automatically extract victim models by giving the monitored cache leakage traces. We have clarified our assumption in Threat Model, Section 3.1. We would also open our source code and dataset later.
>
> **2. The key challenge to attach separable or dilated convs**
>
> The key challenge is to distinguish the separable or dilated convolutions from normal convolutions since they share lots of similarities. It is difficult to differentiate them from the cache access leakage traces. In this paper, we analyzed different leakage patterns between various convolutions and achieved distinguishing them with DNN models. We have emphasized the pattern difference in Figure 2 and the corresponding descriptions.
>
> **3. Is the proposed approach specific to NAS models?**
>
> Our approach is not specific to NAS models. It can be applied to conventional CNN models as well, since the adopted operations are also in our consideration. We focus on NAS because prior works do not work for NAS models. In our revision, we have added some discussions in Section 6 about the generalization of our approach to standard DNN models.
>
> [a] Mengjia Yan,  Christopher W Fletcher,  and Josep Torrellas.   Cache telepathy:  Leveraging shared resource attacks to learn DNN architectures.  InUSENIX Security Symposium, pp. 2003–2020,2020.
>
> [b] Sanghyun Hong,  Michael Davinroy,  Yi ̆gitcan Kaya,  Dana Dachman-Soled,  and Tudor Dumitras ̧.How to 0wn NAS in your spare time.arXiv preprint arXiv:2002.06776, 2020

---

> > ### Comment · Reviewer_KWQS · 2021-11-22
> > **Thanks for the clarification!**
> >
> > My questions are mostly addressed, except I am still not very sure why this technique is specific to NAS but other cannot applied to NAS models. NAS is just a tool to find new models, but the found model has no fundamental difference to other static neural networks.
> >
> > Regardless, this is a very interesting paper, and I recommend accepting this paper.

---

> > > ### Author Response · Authors · 2021-11-23
> > > **Thanks for the comments**
> > >
> > > Great thanks for your kind recommendation of acceptance. Previous model extraction attacks mainly focused on the revealing of normal convolutions and cannot distinguish various kinds of convolutions, i.e., separate or dilated convolutions, so that they cannot extract NAS models that contain  more complex convolutions. Besides, most previous works require the knowledge of model family, which also restrict their application on NAS models that do not have fixed structure template.

---

### Decision · Program_Chairs · 2022-01-20

**Decision:**

Accept (Spotlight)

**Comment:**

All reviewers agree on acceptance and I agree with them. I recommend a spotlight.